# GLOBAL-LOCAL NETWORK FOR
# LEARNING DEPTH WITH VERY SPARSE SUPERVISION

## ABSTRACT

Natural intelligent agents learn to perceive the three dimensional structure of the world without training on large datasets and are unlikely to have the precise equations of projective geometry hard-wired in the brain. Such skill would also be valuable to artificial systems in order to avoid the expensive collection of labeled datasets, as well as tedious tuning required by methods based on multi-view geometry. Inspired by natural agents, who interact with the environment via visual and haptic feedback, this paper explores a new approach to learning depth from images and very sparse depth measurements, just a few pixels per image. To learn from such extremely sparse supervision, we introduce an appropriate inductive bias by designing a specialized global-local network architecture. Experiments on several datasets show that the proposed model can learn monocular dense depth estimation when trained with very sparse ground truth, even a single pixel per image. Moreover, we find that the global parameters extracted by the network are predictive of the metric agent motion.

## 1 INTRODUCTION

Understanding of the three-dimensional (3D) structure of the world is crucial for the functioning of intelligent agents: for instance, it supports path planning and navigation, as well as motion planning and object manipulation. Animals, including humans, obtain such three-dimensional understanding naturally, without any specialized training. By observing the environment and interacting with it, they learn to estimate distances to objects using stereopsis and a variety of monocular cues [1, 2], including motion parallax, perspective, defocus, familiar object sizes. How could artificial systems acquire such spatial awareness?

This question inspired a long line of work on algorithmically extracting 3D structures from their two-dimensional (2D) projections [3]. Classically, multi-view geometry is used to reconstruct the 3D coordinates of points given their corresponding projections in multiple images. These geometric methods, carefully engineered over decades, demonstrate impressive results in a variety of settings and applications [4, 5]. One downside of this class of approaches is that they are very sensitive to hyper-parameters, e.g. camera calibration, which generally require tedious tuning. Additionally, they do not exploit more subtle monocular cues, such as perspective, defocus or familiar object size.

In order to utilize such cues, supervised learning approaches train deep neural networks to predict depth maps from annotated datasets [6]. Since they are not strictly dependent on projective geometry, they can achieve impressive results on sequences with unknown, and potentially varying, camera model and parameters, but require a considerable amount of training data [7]. Collecting such datasets is an expensive process, which is generally repeated every time the application environment changes. To combine the respective advantages of learning and geometry, unsupervised approaches to depth estimation [8, 9] train deep neural networks through projective geometry, eliminating the need to collect large annotated training datasets. These approaches are remarkably successful for some scenes, for instance driving, but have not yet been clearly demonstrated in other environments, such as indoors. Additionally, being based on geometry, they inherit high sensitivity to the camera model and parameters and the need for extensive tuning.

The present work is motivated by the following question: How can 3D perception be learned by an embodied agent without the explicit use of projective geometry or large training datasets? To make the problem tractable, we make two assumptions. First, motivated by the extensive evidence from

psychology and neuroscience on the fundamental importance of motion perception [10, 11, 12, 13], we provide pre-computed optical flow as an input to the depth estimation system. Optical flow estimation can be learned either from synthetic data [14, 15, 16] or from real data in an unsupervised fashion [17, 18]. Second, we explore the idea that perception gets coupled to 3D properties of the world via interaction with the environment. Instead of realistically modeling haptic sensing, we simulate an agent equipped with a low-cost range sensor, similar to a rangefinder or a 2D LiDAR. These provide very sparse depth ground truth, just a few pixels per image. Together with the monocular video, this is the only training signal that the agent receives.

In order to learn from such sparse annotations, we design a lightweight global-local network architecture (see Fig. 1) consisting of two modules – global and local – inspired by camera pose estimation and triangulation in standard geometric pipelines. The global module takes as input two images and the optical flow between them and outputs a compact latent vector of "global parameters", which encodes the observer's motion between the frames and the camera parameters. The local module, consisting of a compact fully convolutional network, combines the "global parameters" with the optical flow field to produce the final depth estimate. Such a global-local decomposition provides our architecture with an inductive bias that is appropriate for learning from very sparse annotations.

Since the method is inspired by learning via interaction, we evaluate it on diverse indoor scenes. We demonstrate that our network learns to estimate depth when provided, at training time, ground-truth for as little as a single pixel per image, i.e., 0.002% of the agent's field of view. We compare against generic deep networks, classic geometry methods, and unsupervised learning approaches. Standard convolutional networks show good results when trained with dense depth ground truth, but their performance degrades dramatically in the very sparse data regime. Both classic and unsupervised learning approaches are generally strong, but exhibit performance drops when calibration parameters are unspecified or dynamically changing. In the sparse data regime, the proposed approach outperforms all baselines thanks to its ability to train with sparse labels and its robustness to variations in the camera parameters.

## 2 RELATED WORK

The problem of recovering the three-dimensional structure of a scene from its two-dimensional projections has been long studied in computer vision [19, 20, 21, 22]. Classic methods are based on multi-view projective geometry [3]. The standard approach is to first find correspondences between images and then use these together with geometric constraints to estimate the camera motion between the images (for instance, with the eight-point algorithm [21]) and the 3D coordinates of the points (e.g., via triangulation [20]). Numerous advanced variations of this basic pipeline have been proposed [23, 24, 4, 5], improving or modifying various its elements. However, key characteristics of these classic methods are that they crucially rely on projective geometry, require laborious hand-engineering, and are not able to exploit non-motion-related depth cues.

To make optimal use of all depth cues, machine learning methods can either be integrated into the classic pipeline, or replace it altogether. The challenge for supervised learning methods is the collection of training data: obtaining ground truth camera poses and geometry for large realistic scenes can be extremely challenging. An alternative is to train on simulated data, but then generalization to diverse real-world scenes can become an issue. Therefore, while supervised learning methods have demonstrated impressive results [6, 7, 25, 26], it is desirable to develop algorithms that function in the absence of large annotated datasets.

Unsupervised (or self-supervised) learning provides an attractive alternative to the label-hungry supervised learning. The dominant approach is inspired by classic 3D reconstruction techniques and makes use of projective geometry and photometric consistency across frames. Existing works use various depth representations for this task: voxel grids [27, 28], point clouds [29, 30], triangular meshes [31] or depth maps [32, 8, 33, 34]. In this work we focus on the depth map representation. Among the methods for learning depth maps, some operate in the stereo setup (given a dataset of images recorded by a stereo pair of cameras) [32, 33], while others address the more challenging monocular setup, where the training data consists of monocular videos with arbitrary camera motions between the frames [8, 34]. Reprojection-based approaches can often yield good results, but they crucially rely on geometric equations and precisely known camera parameters (one notable exception being the recent technical report in [35], which learns the camera parameters automatically). In

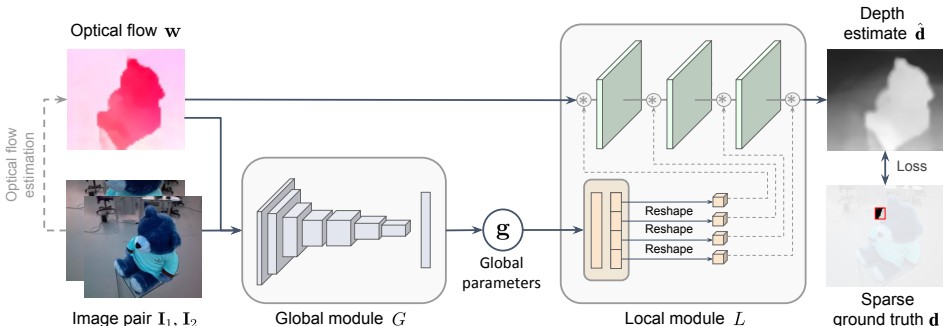

Figure 1: Global-local model architecture. An image pair and an estimated flow field are first fed through the global module that estimates the "global parameters" vector **g**, representing the camera motion. From these global parameters, the local module generates three convolutional filter banks and applies them to the optical flow field. The output of the local module is then processed by a convolution to generate the final depth estimate.

contrast, we do not require knowing the camera parameters in advance and do not rely on projective geometry. Also related to our work are depth completion methods [36], which learn to predict full depth maps given sparse ground-truth annotations. However, while they assume to observe ground-truth also at test time, we use ground-truth annotations only at training time. At test time, we predict depth maps from two images only. Using sparse annotations only for training has been applied mainly to semantic segmentation [37], where additional cues, e.g., object masks, can facilitate learning.

Several works, similar to ours, aim to learn 3D representations without explicitly applying geometric equations [38, 39, 40]. A scene, represented by one or several images, is encoded by a deep network into a latent vector, from which, given a target camera pose, a decoder network can generate new views of the scene. A downside of this technique is that the 3D representation is implicit and therefore cannot be directly used for downstream tasks such as navigation or motion planning. Moreover, at training time it requires knowing camera pose associated with each image. Our method, in contrast, does not require camera poses, and grounds its predictions in the physical world via very sparse depth supervision. This allows us to learn an explicit 3D representation in the form of depth maps.

## 3 METHODOLOGY

Given two monocular RGB images $\mathbf{I}_1$, $\mathbf{I}_2$, with unknown camera parameters and relative pose, as well as the optical flow **w** between them, we aim to estimate a dense depth map corresponding to the first image. We assume to have an artificial agent equipped with a range sensor, which navigates through an indoor environment. By doing so, it collects a training dataset of image pairs, with depth ground truth **d** available only for extremely few pixels. Using this sparsely annotated dataset, we train a deep network $F_{\boldsymbol{\theta}}(\mathbf{I}_1, \mathbf{I}_2, \mathbf{w})$, with parameters $\boldsymbol{\theta}$, that predicts a dense depth map $\hat{\mathbf{d}}$ over the whole image plane. We now describe the network architecture in detail.

### 3.1 MODEL ARCHITECTURE

An overview of the global-local network architecture is provided in Figure 1. The system operates on an image pair $\mathbf{I}_1$, $\mathbf{I}_2$ and the optical flow (dense point correspondences) **w** between them. In this work, we estimate the flow field with an off-the-shelf optical flow estimation algorithm, which is neither trained nor tuned on our data.

The rest of the model is composed of two modules: a global module $G$ that processes the whole image and outputs a compact vector of "global parameters" and a local module $L$ that applies a compact fully convolutional network, conditioned on the global parameters, to the optical flow field. This design is motivated both by classic 3D reconstruction methods and by machine learning considerations. Establishing an analogy with classic pipelines, the global module corresponds to the relative camera pose estimation, while the local module corresponds to triangulation – estimation of depth given the image correspondences and the camera motion. These connections are described in more detail in

the supplement. From the learning point of view, we aim to train a generalizable network with few labels, and therefore need to avoid overfitting. The local module is very compact and operates on a transferable representation – optical flow. The global network is bigger and takes raw images as input, but it communicates with the rest of the model only via the low-dimensional bottleneck of global parameters, which prevents potential overfitting.

The "global module" $G$ is implemented by a convolutional encoder with global average pooling at the end. The network outputs a low-dimensional vector of "global parameters" $\mathbf{g} = G(\mathbf{I}_1, \mathbf{I}_2, \mathbf{w})$. The idea is that the vector represents the motion of the observer, although no explicit supervision is provided to enforce this behavior. While the optical flow alone is in principle sufficient for ego-motion estimation, we also feed the raw image pair to the network to supply it with additional cues.

The "local module" $L$ takes as input the generated global parameters $\mathbf{g}$, as well as the optical flow field. First, the global parameter vector is processed by a linear perceptron that outputs several convolutional filters banks, collectively denoted by $\boldsymbol{\varphi} = LP(\mathbf{g})$. Then, these filter banks are stacked into a small fully convolutional network $C_{\boldsymbol{\varphi}}$ that is applied to the optical flow field. We append two channels of $x$- and $y$- image coordinates to the input $\mathbf{w}$ of $C_{\boldsymbol{\varphi}}$, as in CoordConv [41]. The output of $C_{\boldsymbol{\varphi}}$ is the final depth prediction $\hat{\mathbf{d}} = C_{\boldsymbol{\varphi}}(\mathbf{w})$.

This design of the local module is motivated by classic geometric methods: for estimating the depth of a point it is sufficient to know its displacement between the two images, its image plane coordinates, and the camera motion. In contrast to this standard formulation of triangulation, we intentionally make the receptive field of the network larger than $1 \times 1$ pixel, so that the network has the opportunity to correct for small inaccuracies or outliers in the optical flow input.

## 3.2 Loss function

Similarly to previous work [25, 7], we define the loss on the inverse depth $\hat{\mathbf{z}} \doteq \hat{\mathbf{d}}^{-1}$. This is a common representation in computer vision and robotics [42, 24], which allows to naturally handle points and their uncertainty over a large range of depths. We use the $L_1$ loss on the inverse depth, averaged over the subset $P$ of the pixels that have associated ground truth inverse depth $\mathbf{z}$:

$$\mathcal{L}_{\text{depth}} = \frac{1}{|P|} \sum_{i \in P} |\hat{z}^i - z^i|. \tag{1}$$

To encourage the local smoothness of the predicted depth maps, we add an $L_1$ regularization penalty on the gradient $\nabla \hat{\mathbf{z}} = (\partial_x \hat{\mathbf{z}}, \partial_y \hat{\mathbf{z}})$ of the estimated inverse depth. Similarly to classic structure from motion methods and unsupervised depth learning literature [33], we modulate this penalty according to the image gradients $\partial \mathbf{I}_1$, allowing depth discontinuities to be larger at points with large $\partial \mathbf{I}_1$:

$$\mathcal{L}_{\text{smooth}} = \frac{1}{|\Omega|} \sum_{i \in \Omega} |\partial_x \hat{z}^i| \, e^{-|\partial_x I_1^i|} + |\partial_y \hat{z}^i| \, e^{-|\partial_y I_1^i|}, \tag{2}$$

with $\Omega$ representing the full image plane. The full training loss of our network is a weighted sum of these two terms $\mathcal{L}_{\text{total}} = \lambda_p \mathcal{L}_{\text{depth}} + \lambda_s \mathcal{L}_{\text{smooth}}$.

## 3.3 Model details

In all our experiments the input images have resolution $256 \times 192$ pixels. Unless mentioned otherwise, ground truth depth is provided for a single pixel of each image, but we also experiment with denser ground-truth signals. We use a pre-trained PWC-Net [16] for optical flow estimation.

We use the Leaky ReLU non-linearity in all networks. The global module is implemented by a 5-layer convolutional encoder with the number of channels growing from 16 to 256, with stride 2 in the first 4 layers. The last 256-channel hidden layer is followed by a convolution with 6 channels and global average pooling, resulting in the 6-dimensional predicted global parameter vector $\mathbf{g}$. The local module consists of a single linear perceptron which transforms $\mathbf{g}$ linearly to a 3.9K vector. Empirically, we did not find an advantage in utilizing a multi-layer non-linear perceptron in place of the linear operation. The resulting vector is split into three parts, which are reshaped into filter banks with kernel size $3 \times 3$ and number of output channels 20, 10, 20, respectively. These filter banks, with Leaky ReLUs in between, constitute the compact fully-convolutional depth estimation network

| | Abs-Inv | Abs-Rel | S-RMSE |
|---|---|---|---|
| Formula | $\frac{1}{|\Omega|}\sum_\Omega \left|\frac{1}{d} - \frac{1}{\hat{d}}\right|$ | $\frac{1}{|\Omega|}\sum_\Omega \frac{|d-\hat{d}|}{d}$ | $\sqrt{\frac{1}{|\Omega|}\sum_\Omega E_{\log}^2 - \left(\frac{1}{|\Omega|}\sum_\Omega E_{\log}\right)^2}$ |

Table 1: Metrics for quantitative evaluation of depth accuracy. $d$ is the ground truth depth, $\hat{d}$ is the prediction. For convenience we denote $E_{\log} \doteq \log(d/\hat{d}) = \log d - \log \hat{d}$. For all metrics, lower is better.

$C_\varphi$. The 20-channel network output is then processed by a single $3 \times 3$ convolutional layer to shrink the channels to 1. The design of this compact fully convolutional network has been inspired by the refinement layer used by previous works on supervised depth estimation [25, 7].

We train the model with the Adam optimizer [43] with an initial learning rate of $10^{-4}$ for a total of approximately 94K iterations with a mini-batch size of 16. We apply data augmentation during training. Further details are provided in the supplement.

## 4 EXPERIMENTS

We design our evaluation procedure to address the following questions: (i) Is the proposed global-local architecture at advantage compared to standard deep convolutional networks when learning with sparse depth ground truth? (ii) What is the robustness of our approach to dynamically changing camera parameters? (iii) Do the learned global parameters contain information about the camera motion between the frames? (iv) How does the proposed method compare to a state-of-the-art unsupervised learned depth estimation method. Finally, we validate our design choices with ablation studies. Quantitative comparisons against classic structure from motion methods, as well as additional qualitative experiments, are provided in the supplementary material.

### 4.1 EXPERIMENTAL SETUP

We test the approach on three datasets collected in cluttered indoor environments, either real or simulated. Scenes11 [25] is a large synthetic dataset with randomly generated scenes composed of objects from ShapeNet [44] against diverse backgrounds composed of simple geometric shapes. SUN3D [45] is a large collection of RGB-D indoor videos collected with a Kinect sensor. RGB-D SLAM [46] is another RGB-D dataset collected with Kinect in indoor spaces. For all datasets, we use the splits proposed by [25]. We discard all image pairs with larger relative rotation than $14°$ per axis and larger relative translation than 1 meter per axis. In order to simulate range observations by the agent, we mask out all depth ground truth except for a single pixel (unless mentioned otherwise).

As commonly done in two-view depth estimation methods [25] and in structure-from-motion methods [47], we resolve the inherent scale ambiguity by normalizing the depth values such that the norm of the translation vector between the two views is equal to 1. To quantitatively evaluate the generated depth maps, we adopt three standard error metrics summarized in Table 1.

### 4.2 LEARNING FROM VERY SPARSE GROUND TRUTH

We compare the proposed global-local architecture to strong generic deep models – the encoder-decoder architecture of Eigen et al. [6], the popular fully convolutional architecture DispNet [48], and the multi-scale encoder-decoder of Laina et al. (FCRN) [49]. Note that for a fair comparison with our method, we provide all the baselines with both the image pair and the optical flow field. We additionally tune the models to reach best performance on our task. The details of the tuning process are reported in Section 6.2.1 in the Appendix. We also compare to a reduced-sized DispNet [48] (Small Enc-Dec), that has a number of parameters similar to our model (including both the global and the local module). Its encoder consists of 4 convolutions with $(16, 32, 54, 128)$ filters, with sizes of $(7, 5, 3, 3)$, and stride 2. Its decoder is composed of 4 up-convolutions with $(128, 64, 32, 16)$ filters of size 3 and stride 1. Encoder and decoder layers are connected through skip connections. Finally, we compare against Struct2Depth [9], current state-of-the-art system for unsupervised depth estimation.

As shown in Table 2, our approach outperforms all the baselines in the sparse supervision regime. Specifically, we outperform the architecture of Eigen et al. [6] on average by $53\%$, the architecture of Laina et al. [49] by $22.5\%$, and the fully convolutional DispNet by $20\%$. Indeed, due to

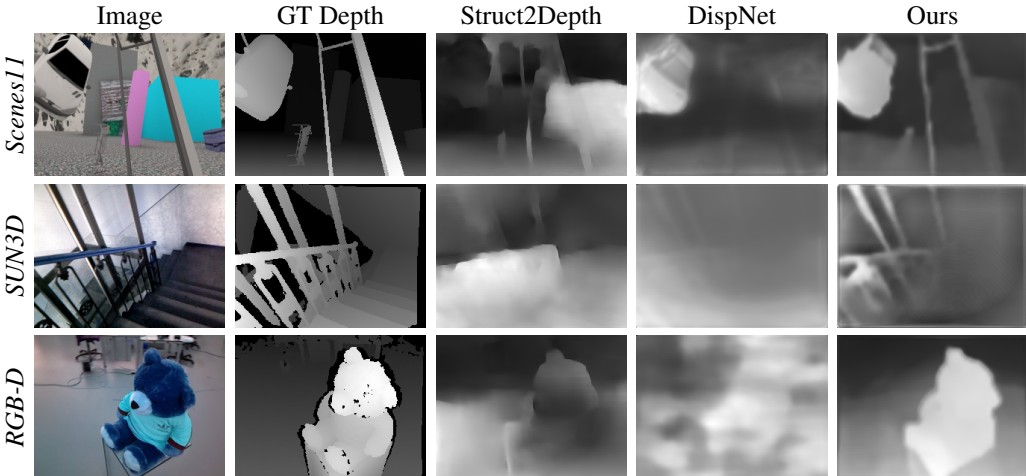

Figure 2: Qualitative comparison of the depth maps generated with the baselines and our approach. Overall, Struc2Depth's predictions are generally poor in homogeneous and repetitive regions, while DispNet tends to over-smooth depth maps. In contrast, our method can predict fine details of the scene geometry.

| Method | Scenes11 | | | SUN3D | | | RGB-D | | |
|--------|---------|---------|--------|---------|---------|--------|---------|---------|--------|
| | Abs-Inv | Abs-Rel | S-RMSE | Abs-Inv | Abs-Rel | S-RMSE | Abs-Inv | Abs-Rel | S-RMSE |
| Eigen [6] | 0.045 | 0.57 | 0.77 | 0.072 | 0.82 | 0.38 | 0.046 | 0.54 | 0.37 |
| DispNet [48] | 0.038 | 0.51 | 0.70 | 0.041 | 0.49 | 0.33 | 0.038 | 0.45 | 0.36 |
| FCRN [49] | 0.041 | 0.52 | 0.74 | 0.047 | 0.44 | 0.30 | 0.042 | 0.45 | 0.35 |
| Small Enc-Dec | 0.046 | 0.66 | 0.83 | 0.064 | 0.73 | 0.45 | 0.049 | 0.58 | 0.46 |
| Struct2Depth [9] | 0.058 | 0.95 | 0.81 | 0.037 | 0.44 | 0.27 | 0.037 | 0.44 | 0.48 |
| Ours | **0.031** | **0.43** | **0.61** | **0.035** | **0.37** | **0.25** | **0.033** | **0.37** | **0.33** |

Table 2: In the sparse training regime, our method can efficiently learn to predict depth from single point supervision, outperforming significantly both standard architectures and unsupervised depth estimation systems. For all error metrics, lower is better.

over-parametrization, these baselines tend to overfit to the training points, failing to generalize to unobserved images and locations.

This is empirically demonstrated in Fig. 3, where we plot the depth loss on training points as a function of the number of iterations. Decreasing the size of the architecture to address overfitting does not however solve the problem: the Small Enc-Dec, with number of parameters similar to our network, achieves poor results, mainly due to its limited capacity.

Our approach also achieves on average $24\%$ better error than the unsupervised depth estimation baseline [9] over all datasets and metrics. Indeed, the considered datasets represent a challenge for geometry-based methods given the presence of large homogeneous regions, occlusions, and small baselines between views, which are typical factors encountered in indoor scenes. Noticeably, the performance of Struct2Depth on the SUN3D dataset is relatively good, boosted by the larger baseline between views and the abundance of features.

Fig. 4 analyzes the performance of our and the DispNet architectures (our strongest baseline) as a function of the number of observed ground-truth pixels per image. Unsurprisingly, both

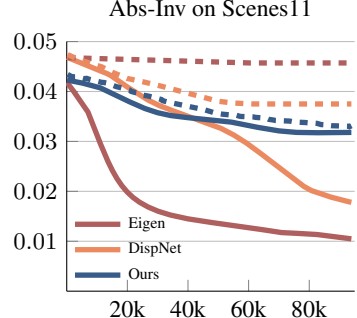

Figure 3: For large networks, the loss on training points (solid lines) is significantly higher than the validation loss (dashed lines). In contrast, our global-local architecture learns generalizible representations.

methods learn to predict accurate depth maps when dense annotations (D) are available. Decreasing the amount of supervision obviously leads to performance drops. However, for our method the error increases on average by only $5\%$ when going to sparser supervision, compared to $12\%$ for the baseline, which leads to a large advantage over the baseline in the single-pixel supervision regime.

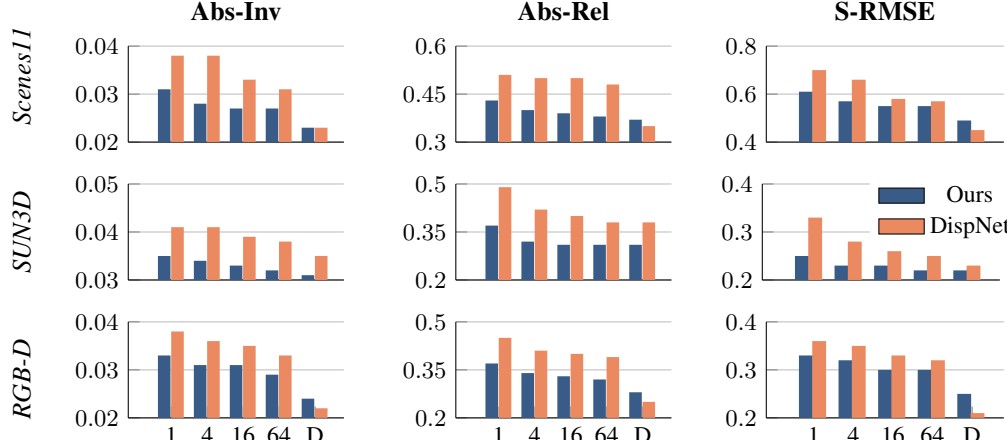

Figure 4: Depth estimation errors with increasing number of training pixels per image and dense supervision (D). When supervision gets sparser, our method's performance degrades more gracefully than the baseline.

This shows that the global-local architecture provides an appropriate inductive bias for learning from extremely sparse depth ground-truth.

### 4.3 Robustness to Dynamically Changing Camera Parameters

In practical applications camera internal parameters, such as focal length, may change through time. Indeed, environmental changes like temperature, humidity and pressure could cause severe variations to their nominal value. Due to these variations, methods based on projective geometry, which are sensitive to the accuracy of calibration parameters, can experience large performance drops. Although the problem could be alleviated by automatic re-calibration, these changes would have to be detected in the first place and would require either collecting multiple views of an object [50, 51] or additional sensing [52].

We empirically study the robustness of our method and the baselines to dynamically changing camera intrinsics. In particular, we randomly change, for each image pair, the horizontal and vertical focal lengths, as well as the center of projection, by up to 20% of their nominal value. The unsupervised depth estimation baseline suffers the most from the uncertainty in the camera intrinsics. Indeed, its estimation error increases on average by 26% with respect to the case in which camera parameters are correctly set. In contrast, as our approach does not explicitly rely on projective geometry, it does not exhibit such sensitivity to the camera parameters. Indeed, it experiences only a small decrease in performance, of approximately 5% with respect to the case where the intrinsics are fixed, since the learning problem becomes more challenging.

| Method | Scenes11 | | | SUN3D | | | RGB-D | | |
|---|---|---|---|---|---|---|---|---|---|
| | Abs-Inv | Abs-Rel | S-RMSE | Abs-Inv | Abs-Rel | S-RMSE | Abs-Inv | Abs-Rel | S-RMSE |
| Struct2Depth [9] | 0.062 | 2.19 | 0.87 | 0.045 | 0.52 | 0.25 | 0.050 | 0.54 | 0.46 |
| DispNet [48] | 0.039 | 0.57 | 0.70 | 0.041 | 0.46 | 0.27 | 0.046 | 0.56 | 0.38 |
| Ours | **0.034** | **0.51** | **0.61** | **0.034** | **0.43** | **0.24** | **0.036** | **0.40** | **0.33** |

Table 3: Depth estimation errors with camera intrinsics varying up to 20% of their nominal value between views. Based on projective geometry, unsupervised methods suffer the most from parameter uncertainty.

### 4.4 Global parameters and the camera motion

According to the intuition behind our model, the global parameters should have information about the observer's ego-motion between the frames, and as such should be related to the actual metric camera motion. Here we study this relation empirically, by training a camera pose predictor on the output of our global module, in supervised fashion. Note that this is done for analysis purposes only, after our full model has been trained: at training time the model has no access to the ground truth camera poses. Specifically, we add a small two-layer MLP with 256 hidden units on top of the global module that is either pre-trained with our method or randomly initialized. We then either train the

| Method | Scenes11 | | SUN3D | | RGB-D | |
|---|---|---|---|---|---|---|
| | rot | trans | rot | trans | rot | trans |
| Scratch-MLP | 1.3 | 74.4 | 3.6 | 55.5 | 5.3 | 78.4 |
| Pretrained-MLP | 0.9 | 26.7 | 2.7 | 32.5 | 4.4 | 51.5 |
| Scratch-Full | **0.7** | 10.3 | 1.8 | 25.0 | **3.2** | 30.5 |
| Pretrained-Full | **0.7** | **9.2** | **1.7** | **22.4** | **3.2** | **28.7** |
| KLT Matlab [25] | 0.9 | 14.6 | 5.9 | 32.3 | 12.8 | 49.6 |
| 8-point FlowFields [25] | 1.3 | 19.4 | 3.7 | 33.3 | 4.7 | 46.1 |

Table 4: Estimation of camera motion based on the global parameters estimated by our model. We initialize the global module either randomly (Scratch) or as trained with our approach (Pretrained). We then append a small MLP and train supervised camera motion prediction by tuning either just the MLP (MLP) or the full network (Full). As a reference, we also report the performance of two classic approaches. We report rotation (rot) and translation (trans) errors in degrees (since the translation vector is normalized to 1, see § 4.1). Lower is better.

full network or only the appended small MLP to predict the camera motion in supervised fashion (details of the training process are provided in the supplement).

Results in Table 4 show that the global parameters indeed contain information about the camera pose. In both training setups pre-trained network substantially outperforms the random initialization: $17\%$ to $64\%$ error reduction across datasets and metrics when only tuning the MLP and up to $11\%$ error reduction when training the full system. Interestingly, our method is also competitive against classic state-of-the-art baselines for motion estimation [25].

### 4.5 ABLATION STUDY

Our architecture is based on several design choices that we now validate through an ablation study. In particular, we ablate the following components: (i) the use of optical flow as an intermediate representation, (ii) the estimation of global variables to generate convolutional filters, (iii) the use of coordinate convolution in the fully convolutional network and (iv) the use of the image pair, in addition to optical flow, for the estimation of global parameters.

The results in Table 5 show that all components are important and some have larger impact than others. The use of optical flow and coordinate convolution are crucial since they both provide essential cues for depth estimation. However, a basic encoder-decoder architecture (*i.e.* without global variables or coordinate convolutions) underperforms even when provided with optical flow. Unsurprisingly, the least important factor is providing the image pair to the global module, since, when camera parameters are fixed, the optical flow is a sufficient statistics of the observer's ego-motion.

| | Abs-Inv | Abs-Rel | S-RMSE |
|---|---|---|---|
| Full Model | **0.033** | **0.43** | **0.61** |
| − Image Pair | 0.033 | 0.45 | 0.62 |
| − CoordConv | 0.038 | 0.52 | 0.71 |
| − Glob. Mod. | 0.041 | 0.55 | 0.73 |
| − Flow | 0.052 | 0.73 | 0.81 |

Table 5: Ablation study on the Scenes11 dataset.

## 5 CONCLUSION

Motivated by the way natural agents learn to predict depth, we propose an approach for training a dense depth estimator from two unconstrained images given only very sparse supervision at training time and without the explicit use of geometry. We show that in cluttered indoor environments our global-local model outperforms state-of-the-art architectures for depth estimation by up to $20\%$ in the sparse data regime.

We see several potential ways for further improving the performance of our approach. The method suffers from outliers in the flow field or when the baseline between views is very small (see Appendix for detailed evaluation). The former problem could be solved by learning depth estimation not only from the sparse ground truth, but also from the photometric reprojection error. This raises the question of how to make use of photometric error without explicit geometric equations. The problem of small baselines could be alleviated by exploiting monocular cues. We see these directions as exciting avenues for future work.

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

## 6 APPENDIX

### 6.1 CONNECTION WITH TWO-VIEW TRIANGULATION

The problem of triangulation consists of computing the 3D coordinates of a point given its (noisy) projections on two or more views and the camera parameters of the views. Hence, it is a geometric problem. Following the usual formalism of homogeneous coordinates [3], the perspective projection of a 3D point $M$ on two cameras with projection matrices $P_1, P_2$ (comprising the intrinsic and extrinsic parameters of both views) is given by $\lambda_1 m_1 = P_1 M$ and $\lambda_2 m_2 = P_2 M$, where $\lambda_1, \lambda_2$ are the projective scaling factors.

Given $P_1, P_2, m_1, m_2$, the linear triangulation algorithm [3, Sec.12.2], which tackles the triangulation problem in its most general setting (projective cameras), computes the 3D point $M$ by minimizing the Rayleigh quotient $\|AM\|/\|M\|$, where $A$ is the matrix

$$A(P_1, P_2, m_1, m_2) = \begin{pmatrix} [m_1]_\times P_1 \\ [m_2]_\times P_2 \end{pmatrix} \tag{3}$$

and $[u]_\times$ is the cross product matrix (such that $[u]_\times v = u \times v$, for all $v$).

In case of multiple point correspondences $\{m_1^i \leftrightarrow m_2^i\}$ for $i = 1, \dots, N$, the camera matrices $P_1, P_2$ appear in the triangulation equations (3) of all of them, and hence, are "global" variables. If the camera matrices are known, then (i) every 3D point $M^i$ can be triangulated independently from the rest, and (ii) the triangulated point is a function of the point correspondences $m_1^i \leftrightarrow m_2^i$.

$$M^i = f(m_1^i, m_2^i; P_1, P_2). \tag{4}$$

This intuition inspired the design of our modular architecture: First, a neural network regresses global variables which depend only on the two views, and then those global variables are used by a local module to generate a fully-convolutional net which transforms point correspondences (optical flow) into depth.

### 6.2 TRAINING PROCESS

We train our model from scratch on the Scenes11 dataset for approximately 150K steps using Adam as optimizer with an initial learning rate of $1e - 4$. We normalize all losses with the number of points used to compute them. The loss weights for depth and smoothness $\lambda_p, \lambda_s$ are 5.0 and 2.0 respectively. To increase generalization, we perform data augmentation at training time by mirroring pairs on the $x$-axis and rotating them 180 degrees, both 50% probability. For a fair comparison, we trained all baselines with exactly the same strategy and hyper-parameters on a desktop PC equipped with an NVIDIA-GeForce 940MX.

For the pose experiments in Sec. 4.4, we trained a 2 hidden layer MLP with 20 nodes and leaky ReLU activation function to predict relative camera motion between frames from the global parameters estimated by our global network. For all datasets and all variations, we trained with an $L_1$ loss between estimated and real camera poses for 50K steps.

#### 6.2.1 TUNING OF BASELINES

To fairly study the sparse training setting, we tuned the baseline to reach the best performance on the sparse training task. First, we changed the input layer of each baseline architecture. Exactly as for ours, the baselines' input consists of the concatenation of the image pair and the optical flow on the last channel. Given the very sparse supervision signal, we noticed that the ReLu activation function generated extremely sparse and noisy gradients. Therefore, we modified the original activation function of DispNet [48], FCRN [49] and Eigen [6] from ReLu to LeakyRelu. This change improved the performance of the baselines of up to $50\%$ on average over metrics.

### 6.3 COMPARISON WITH STRUCTURE FROM MOTION METHODS

Dense Structure from Motion (SfM) methods [24, 23] can recover the depth map of a scene from two or more views using projective geometry [3]. We now compare our global-local architecture to two

SfM baselines proposed by Ummenhofer et al. [25]: one that computes correspondences between images by matching SIFT keypoints (SfM-SIFT) and another that uses optical flow instead [53] (SfM-Flow). Given the correspondences, the essential matrix is estimated with the normalized 8-point algorithm and RANSAC [3], and further refined by minimizing the reprojection error with the *ceres* library. Finally, the depth maps are computed by plane sweep stereo and optimized with the variational approach of Hirshmueller et al. [47].

Results in Table 6 show that our approach achieves on average $63\%$ better error than SfM-SIFT and $43\%$ better error than SfM-Flow over all datasets and metrics. Indeed, the considered datasets represent a challenge for geometry-based methods given the presence of large homogeneous regions, occlusions, and small baselines between views, which are typical factors encountered in indoor scenes. In addition, geometry-based methods are known to be subject to correspondence errors, which the aforementioned factors generally worsen. Nonetheless, our method remains competitive against the geometry-based techniques, outperforming them on two out of three metrics. Furthermore, as we show in the next section, our method is much more robust to errors in the optical flow estimates than classic triangulation.

| Dataset | Scenes11 | | | SUN3D | | | RGB-D | | |
|---|---|---|---|---|---|---|---|---|---|
| | Abs-Inv | Abs-Rel | S-RMSE | Abs-Inv | Abs-Rel | S-RMSE | Abs-Inv | Abs-Rel | S-RMSE |
| Dataset-Mean | 0.069 | 0.771 | 0.940 | 0.081 | 0.730 | 0.378 | 0.062 | 0.695 | 0.475 |
| SfM-SIFT [25] | 0.051 | 1.027 | 0.900 | 0.029 | 0.286 | 0.290 | 0.050 | 0.703 | 0.577 |
| SfM-Flow [25] | 0.038 | 0.776 | 0.793 | **0.029** | **0.297** | 0.284 | 0.045 | 0.613 | 0.548 |
| Ours | **0.033** | **0.43** | **0.61** | 0.045 | 0.48 | **0.23** | **0.040** | **0.44** | **0.33** |

Table 6: Comparison to Structure from Motion (SfM) baselines. Our approach outperforms SfM methods on all datasets and metrics, except for two. This shows that learning from sparse supervision can be competitive with classic geometry-based techniques.

### 6.4 EXPERIMENTS WITH GROUND TRUTH

The main intuition driving the design of our architecture consists of the fact that the relative pose between two images control the conversion of correspondences to depth. Since depth can be computed analytically given the correspondences (in non-degenerate cases), can also our model learn this relation when correspondences are perfect or the relative pose between cameras is given? How does it compare to the triangulation equations in term of performance and ability to handle false correspondences?

To answer these questions, we trained our network with either perfect flow or given pose. As baselines, we used both the simple triangulation equation and the SfM-pipeline presented in Sec. 4.3, but provided with ground-truth camera motion. In addition, we also compare our approach to linear triangulation with the pose predicted by our global parameter network after supervised refinement on ground-truth poses (see Sec. 4.4). The results of this analysis are reported in Table 7.

| Dataset | Scenes11 | | | SUN3D | | | RGB-D | | |
|---|---|---|---|---|---|---|---|---|---|
| | Abs-Inv | Abs-Rel | S-RMSE | Abs-Inv | Abs-Rel | S-RMSE | Abs-Inv | Abs-Rel | S-RMSE |
| Triang.-P. Pose | 0.061 | 0.48 | 0.65 | 0.65 | 0.78 | 0.52 | 0.50 | 0.52 | 0.73 |
| Triang.-GT Pose | 0.020 | 0.25 | 0.48 | 0.14 | 0.33 | 0.44 | 0.091 | 0.36 | 0.49 |
| SfM-GT Pose [25] | 0.023 | 0.35 | 0.62 | **0.020** | **0.22** | 0.24 | **0.026** | **0.34** | 0.398 |
| Ours-GT Flow | **0.015** | **0.24** | **0.46** | 0.026 | 0.26 | **0.20** | 0.040 | 0.35 | **0.32** |
| Ours-GT Pose | 0.021 | 0.28 | 0.54 | 0.038 | 0.37 | 0.32 | 0.043 | 0.44 | 0.38 |
| Ours | 0.033 | 0.43 | 0.61 | 0.045 | 0.48 | 0.23 | 0.040 | 0.44 | 0.33 |

Table 7: Comparison of our approach with different input modalities to triangulation with perfect pose, triangulation with pose estimated by finetuning our global net as in Sec. 4.4 (P. Pose), and the SfM pipeline with ground-truth pose. Although slightly outperformed by the SfM baseline with pose information, our approach is significantly better than naive triangulation on the real datasets, where correspondences are generally very noisy, indicating that our approach learns to filter out these correspondence errors. Providing ground-truth relative pose between images or perfect correspondences generally increases the network performance, showing the ability of our model to learn the relationship between these two modalities in ideal conditions.

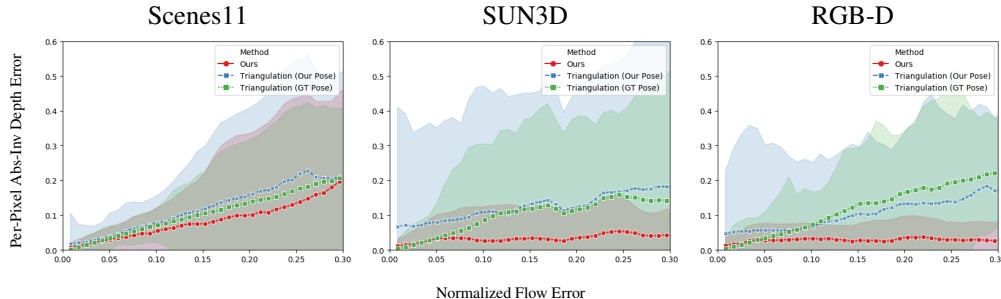

Figure 5: Relation between per pixel flow and depth error for our method and triangulation, either with perfect pose (GT Pose) or with the pose provided by our fine-tuned global network (see Sec. 4.4). Our approach learns to filter out errors in correspondences by exploiting its receptive field larger than one and regularities of those errors in the data.

Performances on Scenes11 show that, when the optical flow generated by PWCNet are very precise, naive triangulation performs very competitively, even outperforming the more complex SfM method. Our network perform comparably to this baseline, showing indeed its ability to learn the mathematical relation between flow and depth. However, triangulation is very sensitive to its inputs: when the precision of the extrinsic parameters or of the correspondences decreases, its performance drastically drops. This can be observed from the results on the SUN3D and RGB-D datasets (Table 7), where the optical flow generated from PWCNet is significantly worse than the one on Scenes11. In contrast, as we show in Fig. 5, our network can cope against these inaccuracies, outperforming triangulation in the real datasets. However, our approach is still not competitive with the SfM pipeline with given extrinsic parameters on the SUN3D and RGB-D datasets: this is an indicator of the difficulty coming from estimating good global and local parameters when both training flows and depths are noisy, but not of our network ability to learn the relationship between these two modalities. Indeed, when the network is provided with perfect correspondences, performance generally increases.

## 6.5 FINE-TUNING OPTICAL FLOW

For all our experiments above, we have always assumed a pre-computed optical flow field is provided as input. This optical flow, generated by the off-the-shelf PWCNet architecture [16], was fixed throughout training. In this section, we study the case when also the parameters of PwCNet are fine-tuned during training with the sparse depth loss. Table 8 shows the results of this evaluation. Interestingly, finetuning the parameters of PWCNet performs worse than fixing them. Such finding can be explained by the fact that the sparse depth loss is not sufficient to train the large number of PWCNet parameters. Indeed, we noticed a decrease in the training error of approximately 10%, indicating an over-fitting to the observed training points. As an additional baseline, we add to PWCNet a small convolutional head to convert flow to depth and train everything with the sparse depth loss. The convolutional head consists of three convolution with (32,16,1) number of filters, stride 1 and filter size 3. The result of this approach, presented in the first row of Table 8, confirms that the sparse loss does not provide enough feedback to fine-tune correspondences.

| Dataset | Scenes11 | | | SUN3D | | | RGB-D | | |
|---|---|---|---|---|---|---|---|---|---|
| | Abs-Inv | Abs-Rel | S-RMSE | Abs-Inv | Abs-Rel | S-RMSE | Abs-Inv | Abs-Rel | S-RMSE |
| PWC [16](*) | 0.046 | 0.63 | 0.81 | 0.081 | 0.87 | 0.37 | 0.047 | 0.51 | 0.42 |
| Ours (finetune PWC) | 0.038 | 0.48 | 0.70 | 0.042 | 0.40 | 0.25 | 0.046 | 0.41 | 0.35 |
| Ours (no finetune) | 0.033 | 0.43 | 0.61 | 0.045 | 0.48 | 0.23 | 0.040 | 0.44 | 0.33 |

Table 8: Fine-tuning the parameters of PWCNet with a very sparse depth loss performs worse than fixing them. (*) Indicates that a small convolutional head has been added to the PWCNet architecture.

## 6.6 QUALITATIVE RESULTS

We show more qualitative results of depth estimation on the test set of our evaluation datasets in Fig. 6, Fig. 7 and Fig. 8. Despite being trained with very sparse supervision, our approach learns to predict smooth depth maps with sharp edges, comparable to the ones an encoder-decoder architecture learns with dense supervision. In contrast, an encoder-decoder architecture fails to learn smooth depths when trained with sparse supervision.

Fig. 9 shows some of the filters produced by the local network to convert optical flow into depth. Since converting flow to depth depends on the relative transformation between the two views, those filters are input-dependent. Generally, filters are different for each image pair. However, when the relative transformation between the input views is similar, filters also tend to be similar (first and second row of Fig. 9). In contrast, when the relative transformation between views is completely different, filters tend to acquire a dissimilar pattern (first and third row of Fig. 9).

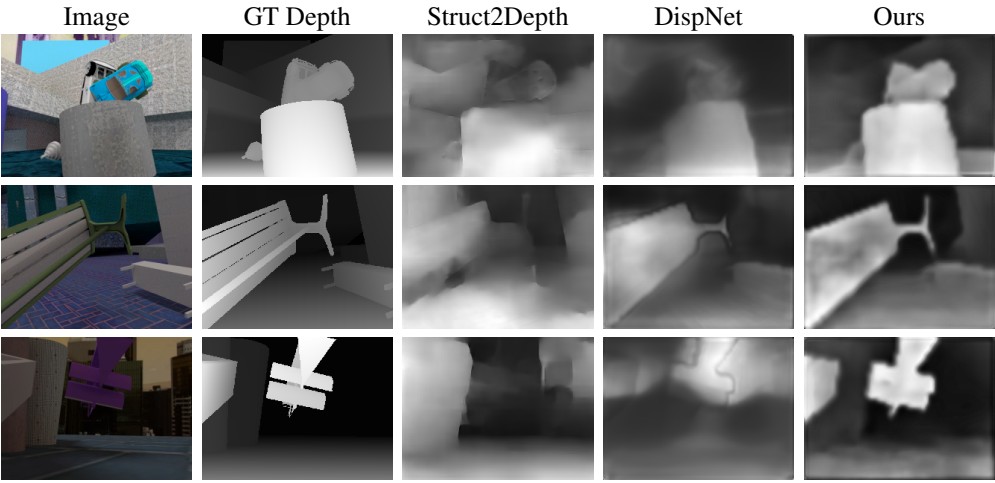

Figure 6: Qualitative results on the simulated Scenes11 dataset. Given the availability of noise-free depth maps for training and high quality optical flow, our approach can learn extremely sharp depth maps, comparable to the ones learned by an encoder-decoder with dense supervision.

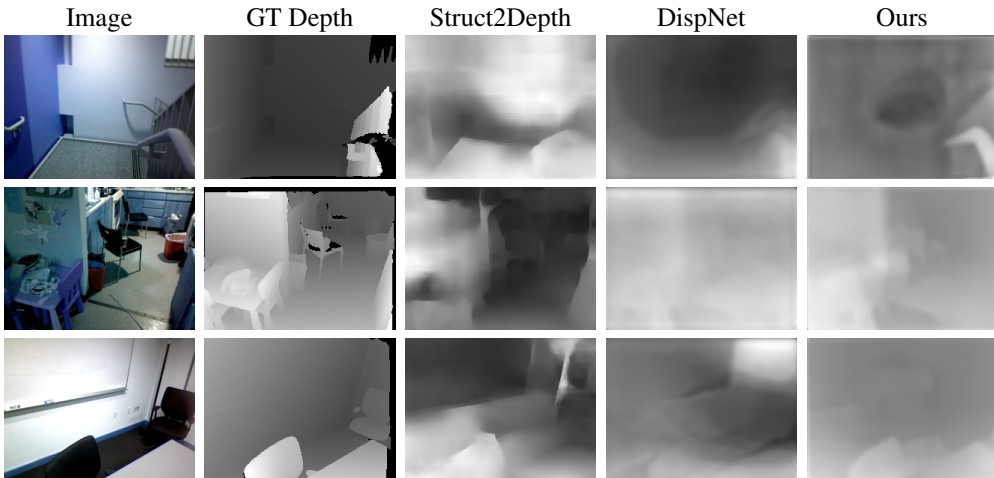

Figure 7: Qualitative results on the SUN3D: Given the very noisy optical flow estimated by PWCNet and the presence of noise in the training depth maps, the performance of all methods drops in this dataset. However, our approach is still able to learn smooth depth maps. Interestingly, our model can pick up details which are not present in the ground-truth depth (top-row).

Image      GT Depth      Struct2Depth      DispNet      Ours

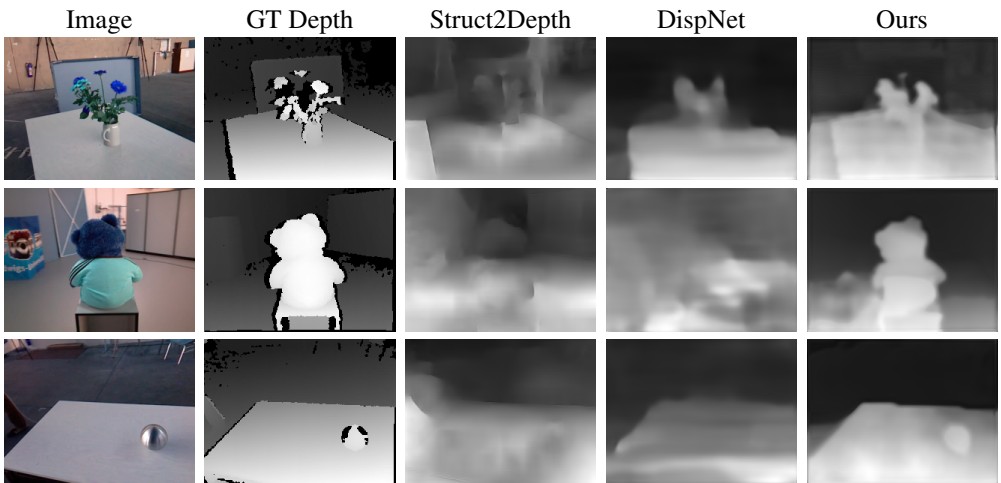

Figure 8: Qualitative results on RGB-D. Also this dataset represents a challenge for all methods, given the large baseline between views, noisy correspondences and noisy training depth maps. Nonetheless, our approach is still able to estimate sharp depth maps, sometimes capturing fine details which even an encoder-decoder trained with dense supervision fails to catch (bottom-row).

Image Pair and Local Network Filters

Figure 9: Local network filters generated by several image pairs. Generally, filters are different for each image pair. However, when the relative transformation between the two views is similar, filters also tend to be similar (first and second row). In contrast, when the relative transformation is completely different, filters tend to have a dissimilar pattern (first and third row).

