# OpenReview forum: "Global-Local Network for Learning Depth with Very Sparse Supervision"
_ICLR.cc/2020/Conference — Reject_

### Official Review · AnonReviewer1 · 2019-10-19
**Official Blind Review #1**

**Rating:** 1

**Review:**

This paper presents a method for learning to predict a depth map given a pair of images. Training is done is an unsupervised way except for a small set of image locations for which the absolute 3D locations are known. The optical flow between the 2 images is computed automatically. A network is trained to predict the camera motion between the 2 images from the images and the optical flow. From this global motion, convolutional filters are predicted, in order to transform the optical flow into a depth map.

The use of image locations with known 3D locations is motivated in the paper to "simulate" the knowledge of depth coming from a haptic system.

I have several concerns about this paper.

* My main concern is that the paper compares against only a few papers (1 from 2014, 1 from  2016, and one more recent from 2019) while the literature is extremely vast, and visually, the results seem far from the state-of-the art. See for example:

Huangying Zhan, Ravi Garg, Chamara Saroj Weerasekera, Kejie Li, Harsh Agarwal, and Ian Reid. Unsupervised learning of monocular depth estimation and visual odometry with deep feature reconstruction. In CVPR, 2018.

Anurag Ranjan, Varun Jampani, Kihwan Kim, Deqing Sun, Jonas Wulff, and Michael J Black. Competitive collaboration: Joint unsupervised learning of depth, camera motion, optical flow and motion segmentation. In CVPR, 2019.

Chaoyang Wang, Jose Miguel Buenaposada, Rui Zhu, and Simon Lucey. Learning depth from monocular videos using
direct methods. In CVPR, 2018.

In fact, these papers have simpler requirements as they do not need 2 images at run-time, nor 3d data at training time. I thought for a moment that the advantage of the paper was to be able to predict an absolute motion and an absolute depth (this is not impossible if the method is able to estimate the scale from known objects, as it is suggested from the introduction). This was however incorrect as the text says in the middle of Section 4.1 " we resolve the inherent scale ambiguity by normalizing the depth values such that the norm of the translation vector between the two views is equal to 1".

* The network used to predict optical flow was trained with a large amount of supervised data. As image matching is the most difficult task, it is difficult to claim that the method is unsupervised.

* The motivation for having sparse measurements is to "simulate" haptic. This is fine for me in principle, however haptic measurements would probably have large errors, while it seems that the experiments use ground truth values for these points.

One minor remark:  End of Section  4.4: What is a "179% error reduction"? How is computed the percentage?


**Experience Assessment:**

I have published one or two papers in this area.

**Review Assessment: Checking Correctness Of Derivations And Theory:**

I carefully checked the derivations and theory.

**Review Assessment: Checking Correctness Of Experiments:**

I carefully checked the experiments.

**Review Assessment: Thoroughness In Paper Reading:**

I read the paper thoroughly.

---

> ### Author Response · Authors · 2019-11-09
> **Answer to AnonReviewer1**
>
> We thank the reviewer for the detailed review and try to address the raised concerns below. We hope these clarifications can help better understanding our experiment design and we are looking forward to hearing back from the reviewer.
>
> We believe that the reviewer might have misunderstood the motivation of our paper, so we briefly restate it here. Existing learning-based depth estimation methods are trained either with a large amount of labeled data or using geometry. Biological agents, instead, acquire 3D awareness by directly interacting with the environment, mainly with haptic feedback. The main research question driving our work is to figure out if artificial agents could also learn from such limited supervision signal,without explicitly using geometry. Investigation of this scientific question may also bring practical benefits: such an approach can be easy to use and robust in applications where appropriate ground truth is available. With this work, we are making the first steps in investigating these questions.
>
> Thus, the main goal of our evaluation is not to thoroughly show the advantage over state-of-the-art methods, but to fairly study the new problem setting. To this end, we compare our proposed architecture to several baselines based on the well-known Eigen and DispNet architectures. Note that our architecture is not provided with any privileged information with respect to these baselines: all of them get access to optical flow, so the comparison is fair. Additionally, we compare to the current state-of-the-art unsupervised depth estimation method [9], which outperforms other methods mentioned by the reviewer (i.e. Huangying et al. CVPR-18, Anurag at al. CVPR-19, Chaoyang et al. CVPR-19)  by a large margin on the KITTI benchmark. We do not compare to the rest of the vast literature on unsupervised depth estimation because those methods have only been applied in a driving setting, which is completely different to our problem setup, and thus each method would require careful and time-consuming tuning to ensure a fair comparison.
>
>
> We respectfully disagree with the statement that “haptic measurements probably have large errors”. Robots (e.g. manipulators) can achieve millimeter accuracies in haptic sensing with the latest technical developments. Haptic sensing has indeed been used both in robotics [1,2] and augmented / virtual reality applications [3]. In addition, neurophysiology studies indicate that such sensing is indeed very fine  for biological agents as well. For example, humans can perceive up to 2.5 degrees in rotation and 3.6mm in translation of the objects they are manipulating [4, Section “Quantifying Haptic Sensitivity”].
>
> Thanks for spotting the typo on “179% error reduction”.  We will correct it in the updated version of the manuscript.
>
> [1]  Marchal-Crespo, Laura, et al. "Haptic error modulation outperforms visual error amplification when learning a modified gait pattern." Frontiers in neuroscience 13 (2019): 61.
> [2] Lew, Thomas, et al. "Contact Inertial Odometry: Collisions are your Friend." arXiv preprint arXiv:1909.00079 (2019).
> [3]  Robles-De-La-Torre, Gabriel. "The importance of the sense of touch in virtual and real environments." Ieee Multimedia 13.3 (2006): 24-30.
> [4]  Henriques, Denise YP, and John F. Soechting. "Approaches to the study of haptic sensing." Journal of Neurophysiology 93.6 (2005): 3036-3043.

---

### Official Review · AnonReviewer3 · 2019-10-23
**Official Blind Review #3**

**Rating:** 6

**Review:**

This paper proposed a novel global-local network, which can be trained with extremely sparse ground truth, to predict dense depth. Though widely applied on the task of segmentation, the use of only uncalibrated input and extremely sparse label for depth estimation is novel. By incorporating optical flow and decoupling global and local modules, this pipeline aligns well with disparity-based geometric methods. The paper is generally well-written and easy to follow.


Here are some concerns:

Some important details are missing in order to reproduce the results. For example, the forward pipelines of the global module and the local module are covered with only few sentences. The authors should expand the section of methodology and give better formulations of their pipeline.

How much does the quality of input optical flow affect the results? Also, the author claimed ‘the network has the opportunity to correct for small inaccuracies or outliners in the optical flow input’, could you show any result related to this claim? (e.g. flawed optical flow but good output)

Can you try to incorporate the optical flow module and fine-tune for better results?

Are there some failure cases, and some visualizations on the filter banks of the local network?

=========================================================
After Rebuttal:

I thank the author for the response.

Since there is a possibly more realistic setting that we train the model with dense depth annotations we currently have and use very sparse annotations to adaptive to new environments where we are hard to gain dense annotations. Also, the optical flow module relies on extra annotations to train. I will keep my origin scores.

**Experience Assessment:**

I have read many papers in this area.

**Review Assessment: Checking Correctness Of Derivations And Theory:**

N/A

**Review Assessment: Checking Correctness Of Experiments:**

I carefully checked the experiments.

**Review Assessment: Thoroughness In Paper Reading:**

I read the paper thoroughly.

---

> ### Author Response · Authors · 2019-11-09
> **Answer to AnonReviewer3**
>
> We thank the reviewer for the positive feedback about our work.
>
> As requested by the reviewer, we have improved the description of our architecture in the methodology section. More details are provided in the Appendix in Section 6.2.  To maximize reproducibility,  we will publish our code upon acceptance.
>
> As the reviewer is interested in the relationship between outliers in optical flow and estimated depth, we invite the reviewer to check out Fig. 5 in the Appendix. This figure experimentally supports the claim that that our approach is more robust to outliers than methods based on projective geometry.
>
> Upon reviewer recommendations, we have added the results of fine-tuning the optical flow module in Table 8 in the Appendix. This methodology gave good results, but did not improve on our previous approach of keeping them fixed throughout training. The reason is that the flow module is too large to be effectively trained with the very sparse depth loss, and ends up overfitting to the observed points. We have validated this claim with additional experiments in which we fine-tune PWCNet end-to-end with only the very sparse depth loss.
>
> In addition, we added a visualization of the filters generated by the local network in Fig.9 of the Appendix. Note that these filters are different for every image pair. However, we discovered that when the relative transformation between images is similar, filters also tend to be similar (first and second row of Fig. 9). In contrast, when the relative transformation between views is completely different, filters tend to acquire a dissimilar pattern (first and third row of Fig. 9).
>
> Finally, we have added a brief summary of failure cases in the conclusion section. In general, our approach suffers from outliers in the optical flow field and by small baseline between views. In future work, we plan to alleviate this problem by fine tuning the optical flow module with the photo-consistency loss and exploiting monocular cues.

---

### Official Review · AnonReviewer2 · 2019-10-24
**Official Blind Review #2**

**Rating:** 3

**Review:**

This paper provides a depth learning architecture (global-local structure) from two images. It claims that SoTA depth can be estimated from the supervision of a very sparse ground truth by leveraging the optical flow information between two images. In the experiments, it shows superior performance than other baseline methods such as Eigen's network and DispNet.

Pros:
1: The paper is well written and motivations are clearly explained.
2: The architecture proposed is reasonable and generate good results, since it accept the ground truth scale from sparse map and relative dense matching cues from optical flow, where implicitly relative camera pose is from global module.


Cons:
1`: It is a fairly standard network design similar to DeMoN[25], where motion network for global pose and local dense network for local matching.

1: The claim of  robustness to camera intrinsics is not solved in principle but due to training using ground truth from multiple dataset . It still suffer from depth motion confusion if there is no ground truth depth guidance when testing.  It is also unfair for comparison of this metric with unsupervised approach where a universal intrinsic is assumed.

2: I think the comparison might not be fair since the baselines are all single image estimation networks, while the approach has two images, where disparities are serving as a strong cue for depth. Other possible  architectures such as flow net, pwc net,  DeMoN[25] and stereo networks such as (gc-net or psm-net) should be considered since these networks are more focus on feature matching.

3:  Even for single image network, eigen's method is not SoTA, the author may consider (1)  as one of the baseline, etc.

(1) "Deeper Depth Prediction with Fully Convolutional Residual Networks"



**Experience Assessment:**

I have published in this field for several years.

**Review Assessment: Checking Correctness Of Derivations And Theory:**

I assessed the sensibility of the derivations and theory.

**Review Assessment: Checking Correctness Of Experiments:**

I carefully checked the experiments.

**Review Assessment: Thoroughness In Paper Reading:**

I read the paper thoroughly.

---

> ### Author Response · Authors · 2019-11-10
> **Answer to AnonReviewer2**
>
> We thank the reviewer for the detailed comments.  We address the raised concerns below. We hope these clarifications help better understanding our contribution and experiment design and we are looking forward to hearing back from the reviewer.
>
> We respectfully disagree that our architecture is very similar to the one proposed in DeMoN [25]. Here are the main differences:
> We predict the convolutional filters in an input dependent way to transform local matches in depth. There is no similar network in the literature to the best of our knowledge.
> We do not explicitly predict camera pose, but a hidden representation of it.
> We do not iteratively transform flow to depth using geometry equations.
>
> Since the reviewer might have slightly misunderstood our problem setup, we briefly summarize it here. We are interested in the problem of learning depth from very sparse supervision signal (available only at training time!), similar to“haptic feedback”, without explicitly using geometry.  Our evaluation was not designed to thoroughly show the advantage over state-of-the-art methods, but to fairly study the new problem setting. In order to do so, we compared to a set of well known architectures for depth estimation, which we tuned to reach the best performance on our task (see Section 6.2.1  in the Appendix). Since our method and the baselines are tuned for the task and trained on exactly the same data, we believe that our evaluation is fair: any advantage our method has is due to the architecture, not the training regime. We have modified section 4.2 to clarify this point.
>
> The experiment with varying intrinsics was designed with the explicit goal of showing that state-of-the art unsupervised depth estimation approaches strongly depend on the camera calibration parameters: when intrinsics are precisely know they can be very good, but are brittle otherwise. Even though concurrent unsupervised approaches can learn intrinsics together with depth [2], they still require intrinsics to be constant over time.  Our method, in contrast, can adapt to instantaneous  changes in camera parameters. Why is this evaluation not fair?
>
> As requested by the reviewer, we added two additional baselines:
> PWC-Net (Table 8, Appendix): We added a convolutional head on top of the off-the-shelf PWC-Net architecture and finetuned the entire system with only the very sparse depth loss. We found its performance to be generally poor, mainly due to overfitting. Details could be found in the Appendix in Table 8.
> The FCRN architecture from [1] (Table 2, main manuscript):  In order to make a fair comparison, we tuned this architecture for our task of learning from sparse supervision. Specifically, we changed its input layer to provide it with the same input as ours:  a pair of frames and the optical flow. We also had to change the activation function of the network from Relu to Leaky Relu to have better performance (details in the appendix in section 6.2.1). We then trained this baseline with exactly the same input as ours, and we found its performance to be lower than our method, but similar to our DispNet baseline [48, manuscript].
>
> [1] Laina, Iro, et al. "Deeper depth prediction with fully convolutional residual networks." 2016 Fourth international conference on 3D vision (3DV). IEEE, 2016.
> [2] Gordon, Ariel, et al. "Depth from Videos in the Wild: Unsupervised Monocular Depth Learning from Unknown Cameras." arXiv preprint arXiv:1904.04998 (2019).

---

### Author Response · Authors · 2019-11-10
**Summary of changes to the manuscript.**

We thank all reviewers for their feedback, which helped us to improve our work. In the following, a summary of the manuscript’s revisions.

Section 4.2: Added details about the training setup for ours and the baselines.
Table 2: Added the baseline requested by Reviewer 2 (FCRN).
Section 5: Added a discussion on failure cases.
Section 6.2.1 (Appendix): Added the details of the baseline tuning process we did to ensure a fair comparison with our method.
Section 6.5 (Appendix): Added an evaluation of fine-tuning correspondences with sparse depth loss, requested by Reviewer 2 and Reviewer 3.
Section 6.6 (Appendix): Added a visualization of the filters learned by the local network, requested by Reviewer 3.

---

### Decision · Program_Chairs · 2019-12-19

**Decision:**

Reject

**Comment:**

This paper proposes a deep network architecture for learning to predict depth from images with sparsely depth-labeled pixels.

This paper was subject to some discussion, since the authors felt that the approach was interesting and the problem-well motivated. Some of the concerns about experimental evaluation (especially from R1) were resolved due the author's rebuttal, but ultimately the reviewers felt the paper was not yet ready for publication.